# Evaluation of microRNA Expression Features in Patients with Various Types of Arterial Damage: Thoracic Aortic Aneurysm and Coronary Atherosclerosis

**DOI:** 10.3390/jpm13071161

**Published:** 2023-07-20

**Authors:** Ange Veroniqe Ngo Bilong Ekedi, Andrey N. Rozhkov, Dmitry Yu. Shchekochikhin, Nina A. Novikova, Philippe Yu. Kopylov, Afina A. Bestavashvili, Tatiana V. Ivanova, Andrey V. Zhelankin, Eduard V. Generozov, Dmitry N. Konanov, Anna S. Akselrod

**Affiliations:** 1Department of Cardiology, Functional and Ultrasound Diagnostics, N.V. Sklifosovsky Institute of Clinical Medicine, I. M. Sechenov First Moscow State Medical University, 119991 Moscow, Russiaagishm@list.ru (D.Y.S.); 7402898@mail.ru (A.S.A.); 2World-Class Research Center “Digital Biodesign and Personalized Healthcare”, I. M. Sechenov First Moscow State Medical University, 119991 Moscow, Russia; 3Federal Research and Clinical Center of Physical-Chemical Medicine, Federal Medical Biological Agency, 119435 Moscow, Russia

**Keywords:** microRNA, thoracic aorta aneurism, atherosclerosis, miR-126, miR-181b-5p, miR-29b-3p, miR-21-5p, miR-92a-3p, HDL

## Abstract

Circulating serum miRNA are increasingly used as biomarkers and potential treatment targets in several clinical scenarios, including cardiovascular diseases. However, the current data on circulating miRNA in thoracic aorta aneurism (TAA) patients are inconclusive. The aim of the present study is to compare the levels of several circulating miRNA in patients with degenerative TAA, coronary artery disease (CAD), and controls for special profile identification. We have identified several candidates for the role of new biomarkers: miR-143-3p, miR-181-5p, miR-126-3p, miR-126-5p, miR-145-5p, miR-150-5p, and miR-195-5p. Materials and methods: Serum samples of 100 patients were analyzed, including 388 TAA patients scheduled for elective surgery, 67 patients with stable CAD and 17 controls, were used for miRNA isolation and identification. Results: More specific for TAA with very high predictive ability in ROC analysis was an increase in the levels of miR-21-5p, miR-29b-5p, miR-126-5p/-3p, miR-181b-5p, and miR-92a-3p, with the latter microRNA being investigated as a novel potential marker of TAA for the first time. Conclusion: TAA and CAD patients demonstrated a significant increase in the levels of circulating miR-126-5p/-3p, miR-181b-5p, and miR-29b-3p. More specific for TAA with very high predictive ability in ROC analysis was an increase in the levels of miR-21-5p, -29b-5p, -126-5p/-3p, 181b-5p, and -92a-3p, with the latter microRNA being investigated as a potential marker of TAA for the first time.

## 1. Introduction

Cardiovascular diseases are the leading cause of morbidity and mortality worldwide, both in developed and developing countries [1,2]. Coronary heart disease and its complications as acute coronary syndromes are the most prevalent disease, followed by cerebrovascular disease and peripheral artery atherosclerosis and aortic diseases. The incidence of thoracic aorta aneurism (TAA) is 5–10 per 100,000 person-years [3]. The TAA patients remain asymptomatic for a long period but could result in a medical emergency. Acute aortic syndrome accounts for high mortality and prolongated hospitalization in successful cases. 

Most cases of TAA are degenerative in their nature and are associated with main cardiovascular risk factors, such as hypertension or hyperlipidemia [4]. However, the direct atherosclerotic etiology is questionable. Moreover, several cases of TAA are attributed to bicuspidal aortic valve disease, genetic arthropathies, or inflammatory diseases.

Research for a novel target for TAA development opens new horizons for treatment and prevention. One potential target is a group of circulating miRNA. MicroRNAs (miRNAs) are a class of small non-coding RNAs that are extraordinarily stable in biofluids and are promising non-invasive biomarkers for many pathological processes. Change in the profile of circulating blood extracellular miRNAs has been detected in a large number of human pathologies, including cardiovascular diseases (CVD) [5,6,7,8]. Data on circulating miRNA in TAA were derived from small case series [9,10] and were inconclusive.

The aim of the present study is to compare the levels of several circulating serum miRNA in patients with degenerative TAA, coronary arteries disease (CAD), and controls for special profile identification. miR-143-3p, miR-181-5p, miR-126-3p, miR-126-5p, miR-145-5p, miR-150-5p, and miR-195-5p were analyzed as potential targets, thus they could reflect atherosclerosis progression and could differentiate between CAD and TAA patients.

## 2. Materials and Methods

### 2.1. Design of the Study

The study included patients from the Departments of Cardiology and Cardiovascular Surgery of Clinical Center of the I. M. Sechenov First Moscow State Medical University (University Clinical Hospital No. 1), Moscow, Russian Federation. Informed voluntary consent was obtained from each participant.

The first cohort of patients was formed from patients whose blood samples were included in the study on coronary atherosclerosis. Detailed characteristics of patients are described in our study [11].

The second cohort of patients for analysis was scheduled for elective thoracic aneurism repair in the cardiovascular surgery department [12]. Patients with acute aortic syndorme, inflammatory aortopathy (e.g., large vessel vasculitis, IgG4-associated diseases), blunt traumatic thoracic aorta injury, chronic aortic dissections, and established genetic aortopathy were excluded for cohort homogeneity, according to the aim of the study. All included patients had degenerative TAA with indications for surgery according to current guidelines [13].

### 2.2. Inclusion Criteria

A written informed consent of the patient to participate in the study.Age 18–80;Availability of blood plasma samples suitable for analysis from the first and second cohorts of patients.

### 2.3. Exclusion Criteria

Any previous heart surgery or coronary interventions (only for first cohort);Severe heart failure (III-IV classes according to the classification of the New York Heart Association NYHA);History of myocardial infarction (only for first cohort);Body mass index 35 or more;The presence of any other severe pathology at the time of the study (including active cancer, impaired liver and kidney function, systemic diseases, inflammatory processes).Refusal of the patient from participation in the study;Psychiatric disorders, including claustrophobia.

### 2.4. Study Duration

The enrollment of patients was carried out from January 2020 to October 2021.

### 2.5. Description of the Intervention

This study unifies two cohorts of patients. The features of the medical intervention of each cohort and the list of miRNAs tested are described in the relevant papers [11,12]. The only medical intervention in this work can be considered blood sampling from patients for subsequent analysis of the level of circulating microRNAs.

### 2.6. Statistical Data Analysis Methods

The Mann–Whitney U-test was used to compare microRNA levels between groups, and the resulting *p*-values were adjusted using the multiple testing Benjamini–Hochberg FDR correction. Moreover, Kruskal–Wallis test *p*-values and the Chi-square test *p*-values were used to show the difference in quantitative and qualitative parameters of the groups, respectively, and all *p*-values were adjusted using the Benjamini–Hochberg FDR correction. Generalized linear models were used to estimate dependency between features (with the Binomial distribution used to describe discrete variables and the Gaussian distribution for continuous variables). Both Spearman’s rank and Perason’s correlation coefficients were used to estimate the correlation between microRNA levels and the selected metadata, including the value of calculated cardiovascular risks via SCORE2, ACC/AHA, and Framingham scales. ROC-AUCs were calculated to demonstrate the predictive power of each particular microRNA.

Statistical analysis was mostly done with Python 3, the Scipy, and statsmodels python libraries used. All plots were built using the matplotlib and seaborn libraries.

## 3. Results

### 3.1. Study Participants

In total, the study included 100 patients (57% women) The characteristics of the studied population are shown in Table 1.

The main distribution of patients was carried out by the presence of thoracic aorta aneurism (n = 38) and coronary arteries atherosclerosis (N = 67). Twenty-two patients with TAA and CAA combination were presented in separate groups. In addition, 17 patients of a cardiology department without CAD and TAA participated in the study (Figure 1).

Due to the presence of a group (n = 22) of comorbid patients, it was decided to conduct comparisons not in pure groups but to take into account CAD and TAA combination. The general cohort of patients with vascular pathology (N = 83) was also considered as a separate group.

### 3.2. Associations of microRNA Levels with Vascular Damage

The relative plasma levels of various miRNAs between the groups had a significant difference in the plasma level of miR 126-5p, miR-126-3p, miR-181b-5p, and miR-29b-3p (Figure 2). All these plasma levels were higher in the vascular damage group and lower in the group with patent arteries and aorta (*p* = 0.035 for miR-126-5p; *p* = 0.018 for miR-126-3p; *p* = 0.026 for miR-181b-5p; *p* = 0.018 for miR-29b-3p). Other miRNA levels did not differ significantly.

A pair-wise comparison was performed using the Mann–Whitney U-test. Relative plasma levels data are presented as mean ± standard deviation. A *p* level <0.05 was considered statistically significant.

ROC analysis demonstrated good predictive efficiency of selected microRNA (Figure 3).

### 3.3. Evaluation of microRNA Expression Profiles in Patients with and without TAA

All the selected microRNAs except for the plasma level of miR-143-3p demonstrated a significant difference in expression between the study groups. The obtained *p*-values are shown in Table 2.

Very good predictive efficiency (AUC > 0.8) in ROC analysis was demonstrated for miR-21-5p, -29b-3p, -92a-3p, -126-3p, -126-5p, and -181b-5p (Figure 4).

### 3.4. Evaluation of microRNA Expression Profiles in Patients with and without Coronary Atherosclerosis

The relative plasma levels of various miRNAs between these groups did not differ significantly, except for the plasma level of miR 126-3p, -21-5p, -29b-3p, and -223-3p (Figure 5). Expression of these miRNAs is lower in CAD group.

A pair-wise comparison was performed using the Mann–Whitney U-test. Relative plasma level data are presented as mean ± standard deviation. A *p* level <0.05 was considered statistically significant.

Moreover, we conducted the ROC analysis of these miRNAs, which demonstrated good and moderate-to-good predictive efficiency of selected microRNA (Figure 6).

### 3.5. Association of miRNA Levels with Clinical and Demographical Characteristics of Patients and Cardiovascular Risks

There were no significant correlations of microRNA levels and calculated CVR values determined by SCORE2, FRS, and ACC/AHA in the studied groups.

Among the other parameters (Table 1), significant correlations (*p* < 0.05) were found in BMI with miR-195-5p; Creatinine with miR-181b-5p, -23a-3p, -150-5p, -92a-3p, -146a-5p, -29b-3p, -223-3p, -21-5p, -126-5p and -126-3p; HDL with miR-21-5p (Figure 7 and Figure 8).

### 3.6. Association of HDL Levels with the Presence of TAA

As part of the analysis of the data obtained, a pronounced association of a lower HDL level with the presence of a thoracic aortic aneurysm was revealed. Adjusted for the use of statins, the significance of this relationship remained at the level of *p* < 0.001; coef = −3.7344 (Figure 9).

## 4. Discussion

The results of the present study indicate changes in the relative plasma levels of several microRNAs in patients with vascular damage. A significant increase in relative plasma levels between the combined TAA + CAD group and the control was demonstrated by miR-126-5p, miR-126-3p, miR-181b-5p, miR- 29b-3p. Interestingly, in a comparative analysis of CAD vs. No CAD, the levels of miR-126-3 panda -29b-5p were lower in the CAD group. 

One of the most studied molecules of various physiological and pathological processes in blood vessels is the miR-126-5p/-3p cluster. The host gene of this miRNA is epidermal growth [14,15] factor-like domain-containing protein 7 (EGFL7) gene, the seventh intron of which encodes pre-miR-126, which is subsequently divided into separate chains miR-126-5p and -3p [16]. The biological role of this microRNA in the framework of vascular damage processes is considered mainly protective. However, the dynamics of these microRNAs in different diseases of the arteries differ [17]. Moreover, the levels depend on the severity of atherosclerosis and the stability of plaques [11,18,19]. In this work, the levels of both miR were elevated in TAA and demonstrate a very high predictive ability in the analysis of ROC. However, in CAD, on the contrary, a relative decrease was demonstrated. Studies on miR-126-5p/-3p levels in TAA are less than in atherosclerosis, but their data are consistent with ours. Thus, in a study by Abu-Halim et al., in patients with aortic dilation with Turner Syndrome, the expression of these microRNAs was significantly higher than in the control group (*p* < 0.0001) [20]. Upregulation of miR-126 was demonstrated in a comparative study on thoracic and abdominal aortic aneurysms. At the same time, there were no significant dynamics of miR-146a levels in TAA in the study, unlike in our work [21]. In Gasiulė et al.’s study, the levels of tissue miR-126-5p/-3p were higher in TAA vs. No TAA group (*p* = 0.0008 for -126-5p and *p* = 0.00002 for -126-3p), but there were no further analyses of the relative plasma levels of these miRNAs [22].

The main pathways of the phase-specific action of miR-126-5p/-3p are suppression of the protein Notch1 inhibitor delta-like 1 homolog (DLK1), inhibiting a number of signaling pathways regulating the proliferation of vascular smooth muscle cells [23] suppression of the VCAM-1 expression, being one of the pro-inflammatory factors, inhibition of the extracellular signal-regulated kinase-mitogen activated protein kinase (MAPK/ERK) pathway also for vascular inflammation reduction [16,24]. Thus, it can be concluded that an increase in miR-126-5p and miR-126-3p, which are atheroprotective molecules, serves as a non-specific marker of both aneurysmal and atherosclerotic arterial damage, with a more pronounced increase in aneurysms. However, additional studies of the dynamics of the levels of these microRNAs are required.

Another molecule of vascular damage in this study is miR-181b-5p, which also showed high predictive ability in ROC analysis. In our previous study, upregulation of this microRNA was noted in atherosclerosis of the coronary arteries, while greater severity was noted in the presence of stable plaques (*p* = 0.0179) in comparison with vulnerable ones. The role of miR-181 b-5p in the processes of vascular damage appears to be atheroprotective [25], acting via NF-kB signaling pathway [26], regulation of the tissue inhibitor of metalloproteinase 3 (TIMP3) expression [27], which differs in different vascular regions. At the same time, the pro-inflammatory effect of this microRNA in atherosclerosis and TAA is also described [28]. However, the latest study was conducted in tissues, not blood plasma. Importin-α3, PI3K, MAPK, KPNA4, and Notch1 can also serve as potential targets for miR-181b-5p [11]. 

The miR-29b-3p molecule, like -181b-5p, is tissue-specific [29], which may affect its plasma levels. The associations of this microRNA with atherosclerosis have been studied relatively widely. The study by Leistner et al. also evaluated the relationship of atherosclerosis-associated microRNAs, including intersecting with our work miR-126-5p/-3p, -181b-5p, -29b-3p, -145-5p, and 92a-3p with the characteristics of atherosclerotic plaque. The levels of microRNA were assessed by qPCR-RT in the aortic bulb and in the coronary venous sinuses with the determination of the transcoronary gradient (TCG) of microRNAs. A significant correlation of the overall degree of coronary atherosclerotic lesion was with miR-126-3p/5p, 145, 29b, and others, but there was no significant correlation with miR-92a. In the study on correlations of microRNA levels and characteristics of atherosclerotic plaque, miR-126-3p/5p, 145-5p, and 29b-3p had a significant inverse correlation with the total volume of fibroatheromas. For miR-126-5p, there was an inverse correlation with thin-cup fibroatheromas (TCFA). In addition, TCG miR-126-3p/5p, -145-5p, and -29b-3p demonstrated a significant distinctive ability to predict the presence of TCFAs (*p* < 0.05). The miR-29b-3p is also supposed to be involved in the calcification of atherosclerotic plaque [30]. The effect of miR-29b-3p on the endothelial function is presumably realized through tumor necrosis factor receptor 1 (TNF-R1), which counteracts the pro-inflammatory effect of tumor necrosis factor alpha (TNFα) [31]. Other potential targets of this microRNA may be protein sprout homolog 1 (SPRY1), a port of MAPK pathway [32], an indirect effect on the activity of the endothelial system NO-synthase is also not excluded [33]. At the same time, the expression of miR-29b-3p in tissues, as well as its circulating plasma levels in patients with TAA, are contradictory [34]. Despite the above effects, a number of studies suggest a provoking effect of miR-29b-3p on the development of TAA [35,36,37]. It is worth noting that in our study, the level of circulating miR-29b-3p in the CAD vs No CAD group was lowered, and in the TAA vs. No TAA group, on the contrary, it was increased and had a high predictive ability in ROC analysis. This is probably due to the tissue specificity of this microRNA, as well as the degree of TAA expression in our sample.

Another microRNA that showed similar dynamics when comparing CAD vs. No CAD and TAA vs. No TAA was miR-21-5p. This microRNA, along with miR-146a-5p, demonstrated regulation in patients with atherosclerosis of both coronary and carotid arteries [38,39,40]. The participation of miR-21-5p in various physiological and pathological processes of the arterial bed has been described quite widely [41]. In atherosclerosis, miR-21-5p participates in the stabilization of the atherosclerotic plaque capsule [42], inhibits the processes of vascular inflammation [43], and with an aneurysm of the abdominal aorta, it slows down its development [44]. At the same time, the expression of miR-21-5p in biological models was higher at TAA [45]. The expression of miR-21-5 in TAA is also increased in humans [22], in particular, against the background of bicuspid aortic valve, and based on the dynamics of the levels of this microRNA, it is already proposed to optimize the treatment tactics in such patients [9]. Potential targets of miR-21-5p are MAPK/ERK pathway by regulation of SPRY1 [41], NF-kB [42,46], and transforming growth factor-β (TGF-β) expression [47]. As in the case of miR29b-3p, in our study, the dynamics of circulating microRNA levels may be related to local expression features. This is confirmed by the results of a study by Parahuleva et al., in which miR-21-5p expression significantly increased in coronary and symptomatic carotid atherosclerotic plaques, while the increase in miR-92a expression was specific only for coronary plaques [48]. At the same time, the dynamics of the atherosclerotic process in unstable angina led to an increase in the expression of miR-21-5p and 92a-3p compared with stable angina, taking into account adjustments for risk factors and medications [49].

In our study, an increase in circulating plasma levels of miR-92a-3p was observed when comparing TAA vs. No TAA and had a very high AUC in the ROC analysis. There was also a correlation with the level of creatinine in the blood (however, there was no correlation with eGFR). This is comparable with the data of Kin et al. In their study, overexpression of miR-92a-3p, as well as miR-21-5p, -29b-3p, -126, -195-5p, and -223, was observed in a slightly lower sample of patients with abdominal aorta aneurism, than that evaluated in our study [50]. However, it is worth noting that the authors noted an increase in hemolysis-specific miR-451a, which could affect the final results. It is worth noting that while there are quite a lot of studies on circulating miR-92a-3p with manifestations of coronary atherosclerosis, we conducted an assessment for TAA for the first time. It is assumed that within the miR-17/92 cluster, this microRNA contributes to the degradation of the extracellular matrix of the thoracic aorta by means of MMP/TIMP pathways and also affects the remodeling of VSMCs by means of TGF-β pathway [51]. 

When analyzing the data of two cohorts of patients, significantly lower HDL levels in patients with TAA attracted attention. Only at the time of preparation of this work for publication, an article by Koba et al. was published, which examines the results of a 26-year follow-up of 95,723 patients, where a decrease in HDL levels was one of the risk factors for the development of rupture or dissection of an aortic aneurysm [52]. Considering the fact that in our study, patients with TAA had a significant severity of aortic lesion, the assessment of the dynamics of HDL levels can play an important role in choosing a treatment strategy for such patients.

### Limitations

The main limitations of this work are insufficiently large samples, as well as the presence of patients with TAA and CAD at the same time. In addition, because of the blood sampling and preparation of plasma samples in different departments, we cannot exclude a certain batch effect. In addition, the presence of only severe forms of thoracic aortic aneurysm requiring surgical treatment can be considered a limitation. Moreover, the study was not prospective, so it is not possible to estimate the association of the levels of the studied microRNAs with the disease outcomes.

## 5. Conclusions

The present study on microRNA expression features patients with various types of arterial damage: Thoracic aortic aneurysm and coronary atherosclerosis showed a significant increase in the levels of circulating miR-126-5p/-3p, -181b-5p, and 29b-3p in patients compared with the control group. In patients with TAA, an increase in the levels of miR-21-5p, -29b-5p, -126-5p/-3p, 181b-5p, and -92a-3p was observed, with the latter microRNA being investigated as a potential marker of TAA for the first time. If the increase in all circulating miR studied was characteristic of TAA requiring surgical treatment, then for CAD, on the contrary, the decrease in miR-126-5p, -21-5p, 29b-3p, and -223-3p levels was observed. It is also worth noting a significant difference in HDL levels, adjusted for statin intake, between patients with CAD and TAA, suggesting that a decrease in HDL levels may be a marker of severe TAA.

## Figures and Tables

**Figure 1 jpm-13-01161-f001:**
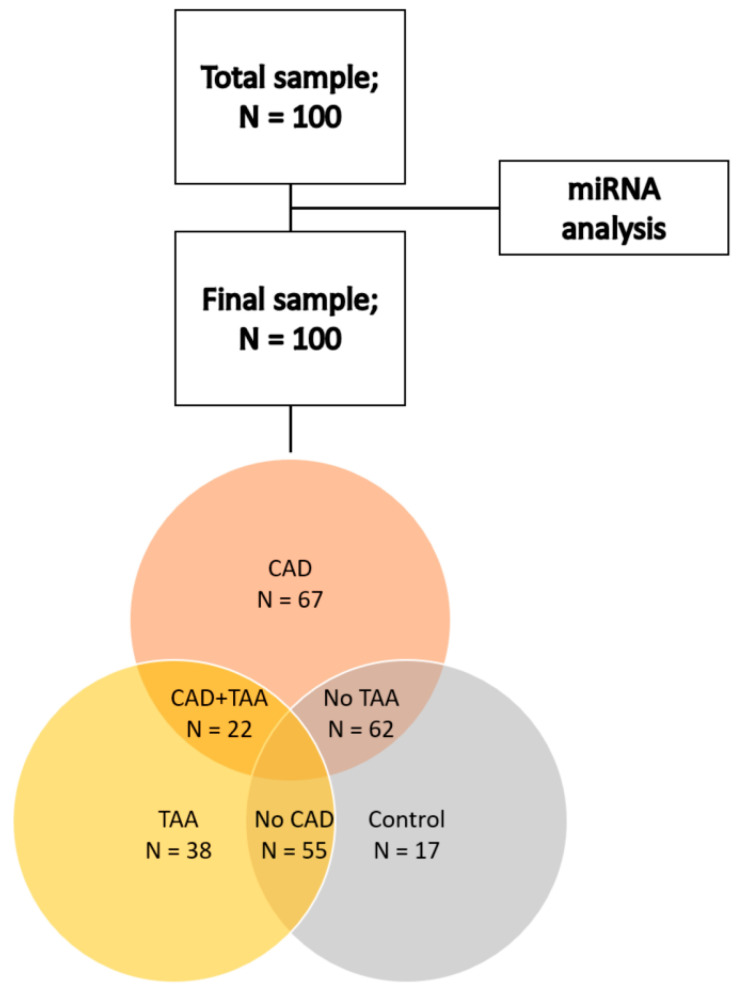
Block diagram of the study.

**Figure 2 jpm-13-01161-f002:**
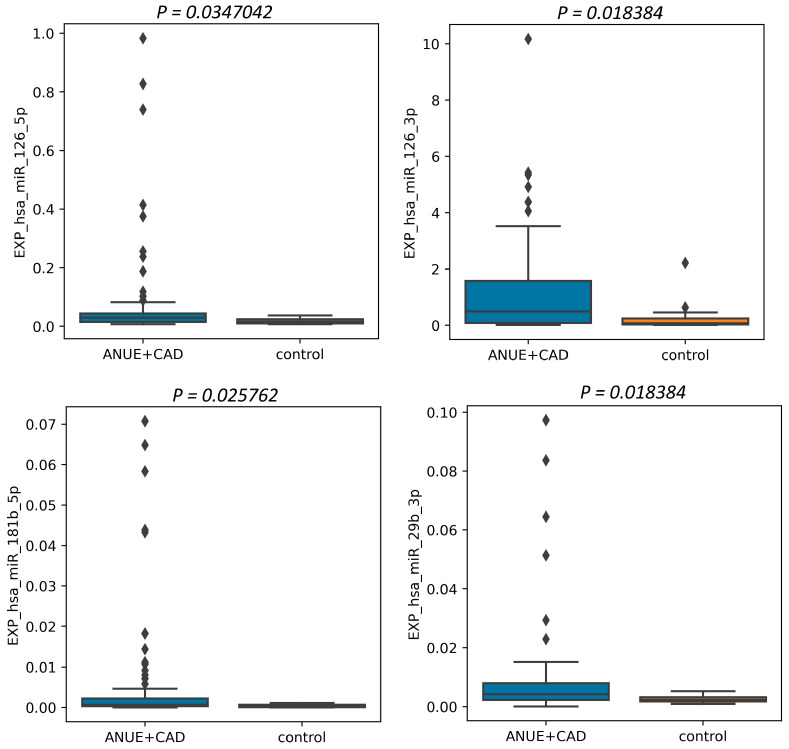
Results of comparison of miRNA relative plasma levels in patients with damaged arteries.

**Figure 3 jpm-13-01161-f003:**
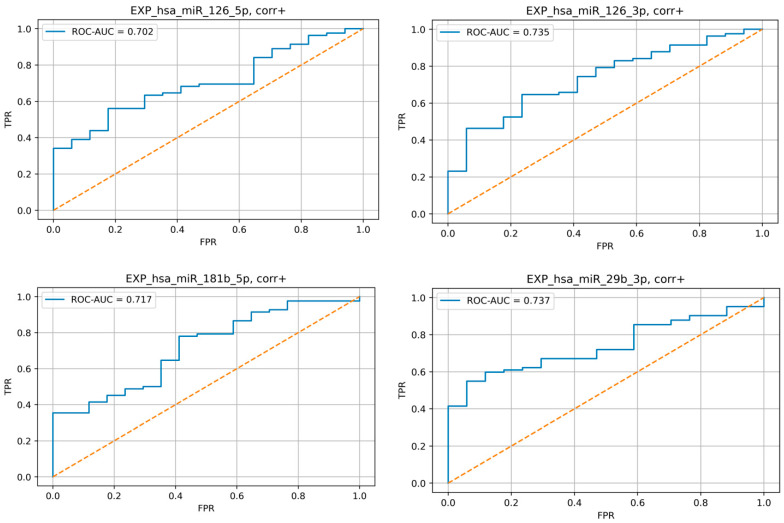
Results of sensitivity and specificity analysis of miR-126-5p, miR-126-3p, miR-181b-5p, and -29b-3p relative plasma levels increase as a marker of arteries damage. ROC curves.

**Figure 4 jpm-13-01161-f004:**
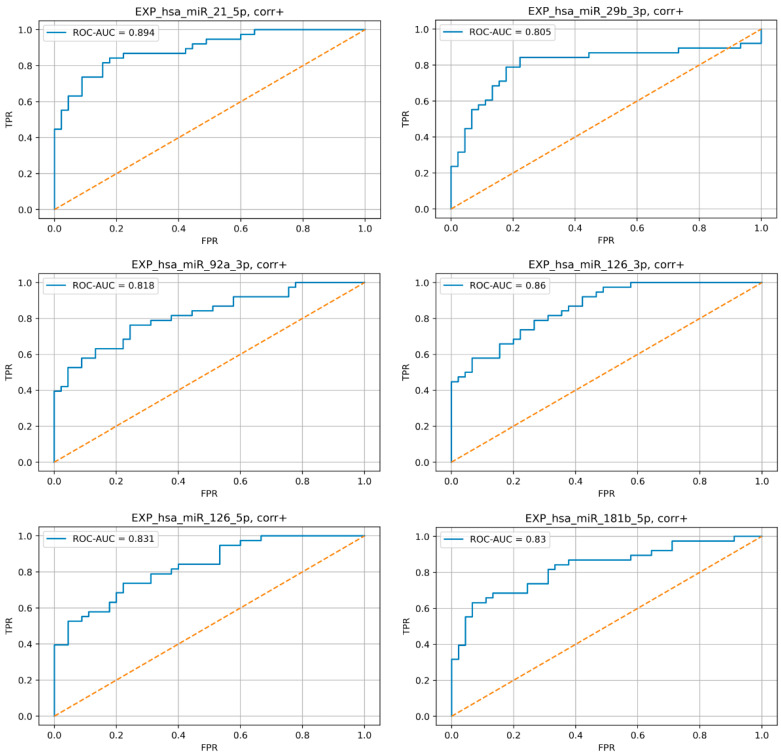
Results of sensitivity and specificity analysis with AUC > 0.8 of the selected miRNA relative plasma levels increase as a marker of TAA. ROC curves.

**Figure 5 jpm-13-01161-f005:**
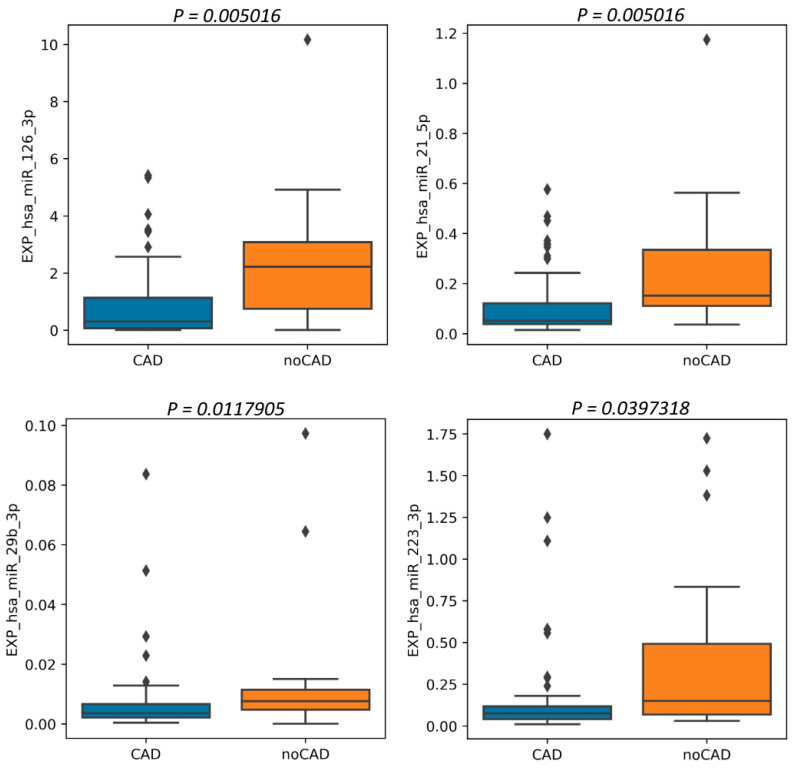
Results of comparison of miRNA relative plasma levels in patients with and without CAD.

**Figure 6 jpm-13-01161-f006:**
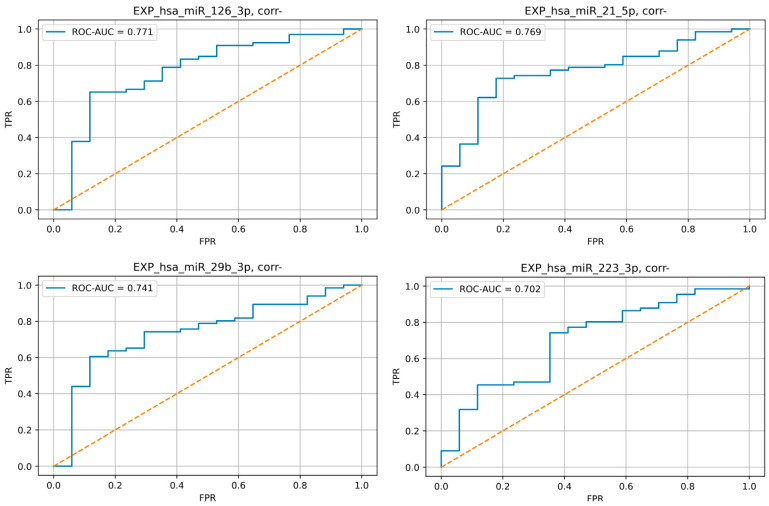
Results of sensitivity and specificity analysis of the selected miRNA relative plasma levels increase as a marker of CAD. ROC curves.

**Figure 7 jpm-13-01161-f007:**
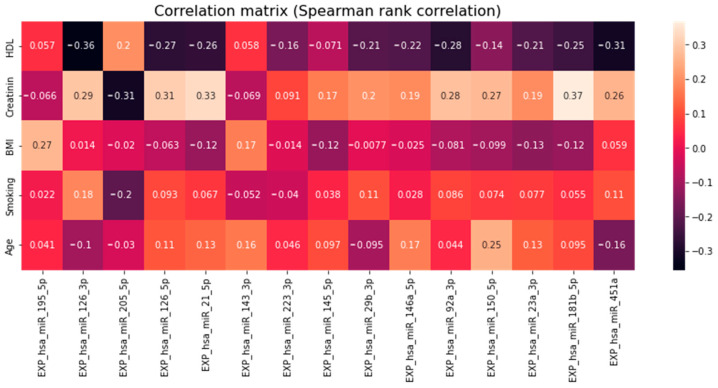
Spearman rank correlation matrix of selected clinical parameters with relative plasma levels of miRNA adjusted for multiple comparisons.

**Figure 8 jpm-13-01161-f008:**
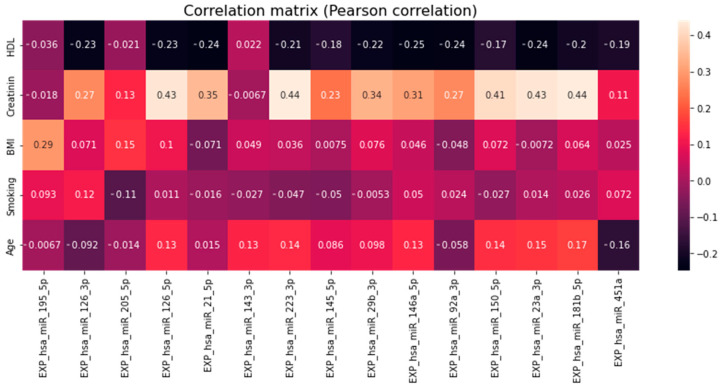
Pearson rank correlation matrix of selected clinical parameters with relative plasma levels of miRNA adjusted for multiple comparisons.

**Figure 9 jpm-13-01161-f009:**
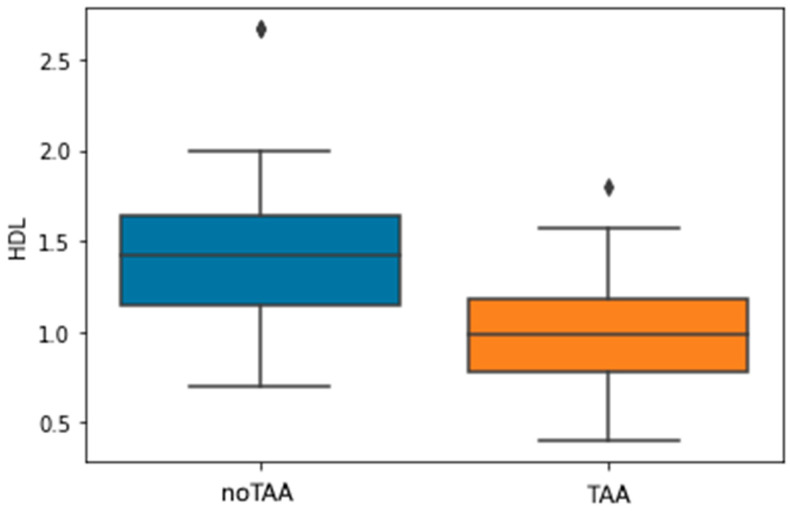
Association of HDL levels with the presence of TAA.

**Table 1 jpm-13-01161-t001:** Characteristics of the population.

Parameter ^1^	Control, *n* = 17		TAA, *n* = 38		CAD, *n* = 67		adj. *p*-Value
	Mean	SD	Mean	SD	Mean	SD	
Age, years	65.0	10.88	63.0	11.93	63.0	9.23	1.0
BMI, kg/m^2^	29.14	04.03	26.79	4.75	27.74	4.3	1.0
Glucose, mmol/L	5.1	0.68	5.45	0.8	5.46	0.83	1.0
Cholesterol, mmol/L	5.68	1.44	4.7	1.14	5.5	1.23	0.112375
Triglycerides, mmol/L	1.43	0.49	1.31	0.94	1.42	0.85	1.0
LDL, mmol/L	3.72	1.38	2.7	01.01	3.24	1.08	0.216216
HDL, mmol/L	1.62	0.26	0.99	0.3	1.21	0.41	5.8 × 10^−5^
VLDL, mmol/L	0.64	0.22	n/a	n/a	0.58	0.43	n/a
Creatinine, µmol/L	84.95	13.37	109.7	20.62	90.1	20.9	0.005963
eGFR_CKD-EPI_, mL/min/1.73 m^2^	66.29	14.6	60.95	18.05	63.0	16.55	1.0
**Parameter**	**Control**		**TAA**		**CAD**		**adj.** ***p*-value**
	**%**		**%**		**%**		
Female Sex	22.2		68.4		45.5		1.0
Smoking	5.6		36.8		25.8		1.0
Type 2 Diabetes mellitus	0.0		15.8		7.6		1.0
Myocardial infarction or PCI	0.0		18.8		9.7		1.0
Stroke	0.0		25.0		8.1		0.949574
Hypertension	88.9		86.8		90.9		1.0
Atrial fibrillation	44.4		28.9		18.2		1.0
Acetylsalicylic acid	44.4		42.1		53.0		1.0
Anticoagulation	33.3		36.8		42.4		1.0
Angiotensin-converting enzyme inhibitors	38.9		21.1		9.1		0.748319
Angiotensin receptor blockers	22.2		39.5		34.8		1.0
Beta-blockers	33.3		18.4		34.8		1.0
Calcium channels blockers	44.4		89.5		72.7		1.0
Hypertension	16.7		39.5		37.9		1.0

^1^ TAA—thoracic aorta aneurism group; CAD—coronary atherosclerosis group; SD—standard deviation; BMI—body mass index; LDL—low-density lipoproteins; VLDL—very-low-density lipoprotein; HDL—high-density lipoproteins; eGFR—estimated glomerular filtration rate; PCI—percutaneous coronary intervention.

**Table 2 jpm-13-01161-t002:** Under- and overexpression of the selected miRNAs in patients with and without TAA.

MicroRNA	*p*-Level
(↓) miR-195-5p	0.000661
(↑) miR-126-5p	0.000001
(↑) miR-223-3p	0.001696
(↑) miR-146a-5p	0.000026
(↑) miR-23a-3p	0.00015
(↑) miR-126-3p	<0.000001
(↑) miR-21-5p	<0.000001
(↑) miR-145-5p	0.001432
(↑) miR-181b-5p	0.000001
(↑) miR-92a-3p	0.000002
(↑) miR-205-5p	0.002194
(↑) miR-143-3p	0.258977
(↑) miR-29b-5p	0.000005
(↑) miR-150-5p	0.009458
(↑) miR-451a	0.00002

## Data Availability

Not applicable.

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
