# Peer review of "Evaluation of microRNA Expression Features in Patients with Various Types of Arterial Damage: Thoracic Aortic Aneurysm and Coronary Atherosclerosis"

_jpm, 2023, doi:10.3390/jpm13071161_

Round 1
Reviewer 1 Report
In this study, the authors compared the levels of circulating serum miRNAs among patients with TAA, CAD, and other diseases. The study identified a few miRNAs with significant differences in their levels in the serum among different patient groups. The authors suggested that some of these miRNAs could serve as biomarkers for TAA. This study is interesting; however, there are some concerns regarding the study design and conclusions.
Main concerns:
1. In Table 1, the authors should list and compare the characteristics of patients in each group instead of listing the entire study population together. This would allow for the review of covariant effects. The authors should also test the covariant effects on the levels of serum miRNAs among the different patient groups.
2. It is not clear how many miRNAs were tested in this study and why the authors selectively presented the results of these miRNAs. Since the authors only compared the serum levels of miRNAs among the TAA, CAD, and 'control' groups, it is not clear whether changes in the levels of some miRNAs might also be observed in other diseases, indicating the specificity of these biomarkers for TAA.
3. When testing multiple miRNAs, the significant p-values should be adjusted using the Bonferroni correction. This correction should also be applied to the section on the 'Association of miRNA levels with clinical and demographic characteristics of patients and 174 cardiovascular risks.'
4. The 'control' group is too small, and the authors also need to specify what disease this group of patients had.
NA
Reviewer 2 Report
Topic is very interesting.
There are numerous errors (type, grammar, nomeclature) that have to be corrected before publication.
Table 1 - missing units.
Figure 1 - needs arythmetic verifiaction.
Adding the conclusion in abstract and in the main text- is needed.
Correlation of miRNA levels with scores SCORE2, FRS, ACC/AHA are mentioned (line 177) but nothing is mentioned in methods about that.
Needs corrections.
Round 2
Reviewer 2 Report
line 21 : check please the meaning of the CAD abbreviation (and delete -s)
table 1 (line 116) the units are still missing (mg/dL? mmol/l?), when you use median it should be accompanied by interquartile rane [25-75], when mean value - SD : decide what you choose
conclusion : blood samples were taken directly before surgical/interventional treatment, so one can say that mRNA profile was typical for devoleped disease, and you say about prediction. You would be able to say about prediction if the blood samples were taken e.g 2 years before event
none
